# Concentration inequalities and optimal number of layers for stochastic deep neural networks

## Abstract

We state concentration and martingale inequalities for the output of the hidden layers of a stochastic deep neural network (SDNN), as well as for the output of the whole SDNN. These results allow us to introduce an expected classifier (EC), and to give probabilistic upper bound for the classification error of the EC. We also state the optimal number of layers for the SDNN via an optimal stopping procedure. We apply our analysis to a stochastic version of a feedforward neural network with ReLU activation function.

## 1 Introduction

Deep neural networks (DNNs) are used extensively in modern statistics and machine learning due to their prediction accuracy; they are applied across many different areas of artificial intelligence, computer vision, speech recognition, and natural language processing (Bahdanau et al., 2016; Hinton et al., 2012; Kalchbrenner & Blunsom, 2013; Krizhevsky et al., 2017; LeCun et al., 2015). Nonetheless, it is common knowledge that the theoretical understanding of many of their properties is still incomplete (for an account on recent results, see Zhang et al. (2018, Section 1)).

In contrast to classical neural networks – which learn mappings from a set of inputs to a set of outputs – stochastic neural networks (SNNs) learn mappings from a set of inputs to a set of probability distributions over the set of outputs. The added complexity introduced by the stochasticity exacerbates the study of their theoretical properties. SNNs were introduced by Wong (1991) to generalize Hopfield networks (Hopfield, 1982). Following Jospin et al. (2022), let us briefly describe SNNs; they are a particular type of artificial neural networks (ANNs). The goal of ANNs is to represent an arbitrary function $y = \Phi(x)$. Traditional ANNs are built using one input layer $\nu^{(0)}(x) = x$, a sequence of of hidden layers $(\nu^{(l)}(x))_{l=1}^{L-1}$, and an output layer $\nu^{(L)}(x)$, for some $L \in \mathbb{N}$. In the simplest architecture of feedforward neural networks, each layer is represented by a linear transformation followed by a nonlinear operation $\sigma^{(l)}$ called *activation function*,

$$
\begin{aligned}
\nu^{(0)}(x) &= x, \\
\nu^{(l)}(x) &= \sigma^{(l)}(A^{(l)}\nu^{(l-1)}(x) + b^{(l)}), \quad l \in \{1, \ldots, L\}, \\
\nu^{(L)}(x) &= y.
\end{aligned}
$$

Let $A = \{A^{(1)}, \ldots, A^{(L)}\}$ and $b = \{b^{(1)}, \ldots, b^{(L)}\}$. Then, $\theta = (A, b)$ represents the parameters of the network, where the $A^{(l)}$'s are weight matrices, and the $b^{(l)}$'s are bias vectors; call $\Theta$ the space $\theta$ belongs to. Deep learning is the process of regressing parameters $\theta$ from a training data $D$ composed of inputs $x$ and their corresponding labels $y$. Stochastic neural networks are a type of ANN built by introducing stochastic components to the network. This is achieved by giving the network either a stochastic activation or stochastic weights to simulate multiple possible models with their associated probability distribution. This can be summarized as

$$
\begin{aligned}
\theta &\sim p(\theta), \\
y &= \Phi_\theta(x) + \varepsilon,
\end{aligned}
$$

where $\Phi$ depends on $\theta$ to highlight the stochastic nature of the neural network, $p$ is the density of a probability measure on $\Theta$ with respect to some $\sigma$-finite dominating measure $\mu$, and $\varepsilon$ represents random noise to account for the fact that function $\Phi_\theta$ is just an approximation. A well known example of SNN are Bayesian neural networks (BNNs), that are ANNs trained using a Bayesian approach (Goan & Fookes, 2020; Jospin et al., 2022; Lampinen & Vehtari, 2001; Titterington, 2004; Wang & Yeung, 2021).

Stochastic neural networks have several advantages with respect to their deterministic counterpart: they have greater expressive power as they allow for multi-modal mappings, and the stochasticity can be seen as added regularization (Lee et al., 2017). In Jospin et al. (2022), the authors point out that the main goal of using SNNs is to obtain a better idea of the uncertainty associated with the underlying process.

The theoretical features of stochastic deep neural networks (SDNNs) have been the object of study of many recent papers. In Merkh & Montúfar (2019), the authors study universal approximation properties of deep belief networks, a class of stochastic feedforward networks that were pivotal in the resurgence of deep learning. A key observation is that, for a finite number of inputs and outputs, the number of maps in the stochastic setting is infinite, unlike in the deterministic framework. This points to the massive increase in the approximation power of SDNNs. In De Bie et al. (2019), SDNNs are framed as learning measures (LMs); deep architectures are introduced to address issues that arise in LMs such as permutation invariance or equivariance and variation in weights. Training stochastic feedforward networks is significantly more challenging. To address this issue, Tang & Salakhutdinov (2013) proposed a stochastic feedforward network with hidden layers composed of both deterministic and stochastic variables with a novel generalized EM training procedure that allows the user to efficiently learn complicated conditional distributions.

In this work, we first investigate concentration inequalities for the hidden layers of a SDNN.

**Proposition 1.** *Let $(\nu^{(l)}(X))_{l=1}^L$ denote the sequence of outputs of the hidden layers of a generic SDNN. If*

- *either $\nu^{(l)}(X)$ is bounded*

- *or the sequence of centered outputs $(\nu^{(l)}(X) - \mathbb{E}[\nu^{(l)}(X)])_{l=1}^L$ is a smooth weak martingale,*

*then $\nu^{(l)}(X)$ concentrates around its expected value.*

Proposition 1 is stated informally. The first bullet point is proven in Corollary 5 in section 2.1, and the second one in Corollary 7 in section 2.2, where we also state conditions that allow us verify the martingale hypothesis.

Furthermore, we prove the following bound on the classification error of the expected classifier.

**Proposition 2.** *If the output $s(\nu(X))$ of the score function that we choose for our analysis is bounded, then the score concentrates around its expected value $\mathbb{E}[s(\nu(X))]$. The classification error obtained using a classifier based on $\mathbb{E}[s(\nu(X))]$ is small.*

Proposition 2 is informal as well. The first part is proven in Proposition 12, and the second one in Proposition 13, both in section 2.3.

In addition, we provide a procedure to find the optimal number of layers.

**Proposition 3.** *A backward induction approach to an optimal stopping problem can be used to find the number of layers of a generic SDNN that strikes the perfect balance between accuracy of the analysis and computational cost.*

Proposition 3, which is informal and that is built on results from Peskir & Shiryaev (2006, Section 1.2), is proven in Theorem 15 in section 3.

Finally, we apply our findings to a stochastic version of the feedforward neural network with ReLU activation studied in Zhang et al. (2018).

## 1.1 Previous work

Concentration inequalities for SDNNs have been studied in the context of PAC-Bayes bounds and stochastic gradient descent (SGD) solutions. In particular, in Huang et al. (2020) the authors propose the Kronecker Flow, an invertible transformation-based method that generalizes the Kronecker product to a nonlinear formulation, and uses this construction to tighten PAC-Bayesian bounds. They show that the KL divergence in the PAC-Bayes bound can be estimated with high probability (they give a Hoeffding-type concentration result), and demonstrate the generalization gap can be further reduced and explained by leveraging structure in parameter space. In Zhu et al. (2022) the authors study the concentration property of SGD solutions. They consider a very rich class of gradient noise – not imposing restrictive requirements such as boundedness or sub-Gaussianity – where only finitely-many moments are required, thus allowing heavy-tailed noises. In the present work, we focus on concentration inequalities (of the Chernoff type) for the output of the hidden layers and of the whole SDNN.

The other focus of our work is finding the number of layers of a generic SDNN that strikes a balance between computational cost and accuracy of the analysis. A similar problem was studied in Trelin & Prochazka (2019). There, the authors present Binary Stochastic Filtering (BSF), an algorithm for feature selection and neuron pruning in a special version of the classical deep neural network structure where stochastic neurons are mixed with deterministic ones. To the best of our knowledge, the present paper is the first to present an optimal stopping procedure to select the number of layers in a generic SDNN.

## 1.2 Contributions

Inspired by Merkh & Montúfar (2019), this paper studies universal properties of stochastic deep neural networks. Our first goal is to give concentration inequalities for the outputs of the hidden layers of a generic SDNN. This problem has already been studied for particular types of SDNNs in Garnier & Langhendries (2021); Ost & Reynaud-Bouret (2020). In the former, the authors establish a framework for modeling non-causal random fields and prove a Hoeffding-type concentration inequality; it is especially important because it can be applied to the field of Natural Language Processing (NLP). In Ost & Reynaud-Bouret (2020), the authors introduce a new stochastic model of infinite neuronal networks, for which they establish sharp oracle inequalities for Lasso methods and restricted eigenvalue properties for the associated Gram matrix with high probability. Their results hold even if the network is only partially observed; their use of stochastic chains inspired the martingale inequalities of section 2.2. Our results are very general because they require little mathematical structure: instead of focusing on non-causal random fields or on infinite neuronal networks, we only require that the output $\nu^{(l)}(X)$ of the $l$-th hidden layer of the SDNN is a random vector. The $\nu^{(l)}(X)$'s can be correlated with each other, and also have different dimensions. The relevance of our findings is given by the lack of specific hypotheses: they apply to any SDNN. In Theorem 4 we show that, under a reasonable assumption, the output $\nu^{(1)}(X)$ of the first layer $\nu^{(1)}$ of SDNN $\nu$ concentrates around its expected value. In Corollary 5 we show that this is true for the outputs of all the layers, so in turn it is also true for the output of the neural network. In Corollary 7 we show how, if the neural network is a weak martingale, then under a weaker assumption than the one in Theorem 4, the output $\nu^{(l)}(X)$ of the $l$-th layer $\nu^{(l)}$ of SDNN $\nu$ concentrates around its expected value. In Theorem 9 we give a sufficient condition for our neural network to be a weak martingale. In Proposition 12 we show that, under a mild assumption, the classifier of our stochastic neural network concentrates around its expected value. This gives us an *expected decision boundary* (EDB). In Proposition 13 we give a probabilistic bound to the classification error of the classifier based on the EDB, that we call the *expected classifier* (EC).

We then turn our attention to the number of layers to select. When specifying the structure of a generic DNN $\nu$, we face a trade-off between accuracy and computational efficiency. One of the main drivers of this trade-off is the number of layers: more layers may yield more accurate analysis, but they also training more computationally intensive. To solve this problem, in Liu & Deng (2018) the authors introduce a new type of DNN called a Dynamic Deep Neural Network (D$^2$NN) that selects a subset of neurons to execute computations. Given an input, only a subset of D$^2$NN neurons are executed, and the particular subset is determined by the D$^2$NN itself. In Shen et al. (2021), the authors introduce a (deterministic) DNN called Floor-Exponential-Step (FLES) network that only requires three hidden layers to achieve super

approximation power. In Sabuncu (2020), the author points out that the problem of selecting the correct number of layers has been studied also when stochasticity is featured in the architecture design of the neural network. In Huang et al. (2016), for example, the authors study DNNs with stochastic depth: the number of layers is chosen randomly, while in Li & Talwalkar (2020); Xie et al. (2020) the authors use a random search algorithm to optimize over the space of neural network architectures. Given a generic SDNN, we adopt an optimal stopping approach to the problem. In Theorem 15, we find the number $\tau_1^L$ of layers for the SDNN that optimizes the trade-off between accuracy and computational cost. To the best of our knowledge, this is the first time such a problem is studied via an optimal stopping procedure for generic SDNNs.

Finally, we apply our findings to a stochastic feedforward neural networks with ReLU activation (SFNNRA). We generalize the setup in Zhang et al. (2018) – that builds a bridge between tropical geometry and deep neural networks – by letting the weight matrix and the bias vector in every layer be stochastic and possibly correlated with the ones in the previous layers. Although the tropical geometry of deep neural networks has been further investigated (Alfarra et al., 2021; Maragos et al., 2021), this is the first time probabilistic results are presented within the framework introduced in Zhang et al. (2018). They are extremely significant: in Proposition 16 we show that – because the assumption of Theorem 4 is easily satisfied in the context of SFNNRAs – the output of every layer concentrates around its expected value. In addition, in Proposition 19 we show that the upper bounds to the number of connected regions with value above and below the decision boundary in the SFNNRA concentrate too around their expected value. In the future, as more general results for SDNNs like the ones presented in this paper are discovered, applying such results to SFNNRAs will improve our understanding of their theoretical properties.

The paper is organized as follows. In section 2 we provide concentration inequalities for the hidden layers of a SDNN and for its output. In section 3 we provide results for the optimal number of layers. In section 4 we apply our results to a stochastic version of the feedforward neural networks with ReLU activation presented in Zhang et al. (2018). Section 5 is a discussion. To make the paper self-contained we provide background on sub-Gaussian random variables and norm-sub-Gaussian random vectors in appendix A, on martingales and filtrations in appendix B, and on tropical algebra in appendix C. We prove our results in appendix D.

### 1.3 Notation

We introduce the notation for deep neural networks (DNNs) that we use throughout the paper. An $L$-layered DNN is a map $\nu : \mathbb{R}^d \to \mathbb{R}^p$. We denote the *width* of the $l$-th layer, that is, the number of nodes of the $l$-th layer, by $n_l$, $l \in \{0, \dots, L\}$, where $n_0 = d$ and $n_L = p$, the dimensions of the input and the output of the network, respectively. The output of the $l$-th layer is given by $\nu^{(l)} : \mathbb{R}^{n_{l-1}} \to \mathbb{R}^{n_l}$. We assume for convenience that $\nu^{(0)}(x) \equiv x$. The final output $\nu(x) \equiv \nu^{(L)}(x)$ of the neural network is fed into a *score function* $s : \mathbb{R}^p \to \mathbb{R}^m$ that is application specific. We call $\mathcal{X}$ the space of inputs and $\mathcal{Y}$ the space of responses. We assume we collect data $(X, Y) \sim \mathscr{D}$, a distribution on $\mathcal{X} \times \mathcal{Y}$. In a SDNN, $(\nu^{(l)}(X))_{l=0}^L$ is a sequence of random vectors having possibly different dimensions and being possibly correlated. Notice that $\nu^{(l)}(x)$ is the realization of random vector $\nu^{(l)}(X) : \Omega \to \mathbb{R}^{n_l}$ whose elements may be correlated. Throughout this paper, we assume that $\nu^{(l)}(X)$ has a finite first moment, for all $l \in \{1, \dots, L\}$.

## 2 Concentration inequalities

In this section we derive concentration inequalities for the hidden layers of a generic SDNN. The results in this section are direct applications of standard tools in high-dimensional probability.

### 2.1 Norm-sub-Gaussian-based concentration inequalities

The results in this section rely on properties of sub-Gaussian random variables and norm-sub-Gaussian random vectors, see appendix A for details. Let $\| \cdot \|_2$ denote the Euclidean norm; the following is the first result.

**Theorem 4.** *Suppose that $\nu^{(1)}(X)$ is bounded so that $\|\nu^{(1)}(X)\|_2 \leq \xi_{(1)}$, for some $\xi_{(1)} \in \mathbb{R}$. Then, for all $t \in \mathbb{R}$,*

$$P_{(X,Y)\sim\mathscr{D}}\left(\|\nu^{(1)}(X) - \mathbb{E}_{(X,Y)\sim\mathscr{D}}\left[\nu^{(1)}(X)\right]\|_2 \geq t\right) \leq 2\exp\left(-\frac{t^2}{2\xi_{(1)}^2}\right). \tag{1}$$

Theorem 4 tells us that the output $\nu^{(1)}(X)$ of the first layer of our SDNN concentrates around its expected value $\mathbb{E}_{(X,Y)\sim\mathscr{D}}[\nu^{(1)}(X)]$. The assumption that $\nu^{(1)}(X)$ is bounded is mild: it can be interpreted as a safety check to ensure that output $\nu^{(1)}(X)$ does not take on values that are too extreme.

We can use Theorem 4 to find a concentration inequality for the second layer of our neural network. In particular, if $\|\nu^{(2)}(X)\|_2 \leq \xi_{(2)}$, for some $\xi_{(2)} \in \mathbb{R}$ possibly different from $\xi_{(1)}$, we have that, for all $t \in \mathbb{R}$,

$$P_{(X,Y)\sim\mathscr{D}}\left(\|\nu^{(2)}(X) - \mathbb{E}_{(X,Y)\sim\mathscr{D}}\left[\nu^{(2)}(X)\right]\|_2 \geq t\right) \leq 2\exp\left(-\frac{t^2}{2\xi_{(2)}^2}\right). \tag{2}$$

This tells us that the output $\nu^{(2)}(X)$ of the second layer of our SDNN concentrates around its expected value $\mathbb{E}[\nu^{(2)}(X)]$.

The following corollary tells us that the result in Theorem 4 holds for every layer of our SDNN.

**Corollary 5.** *Pick any $l \in \{1, \ldots, L\}$. Suppose that $\nu^{(l)}(X)$ is bounded so that $\|\nu^{(l)}(X)\|_2 \leq \xi_{(l)}$, for some $\xi_{(l)} \in \mathbb{R}$. Then, for all $t \in \mathbb{R}$,*

$$P_{(X,Y)\sim\mathscr{D}}\left(\|\nu^{(l)}(X) - \mathbb{E}_{(X,Y)\sim\mathscr{D}}\left[\nu^{(l)}(X)\right]\|_2 \geq t\right) \leq 2\exp\left(-\frac{t^2}{2\xi_{(l)}^2}\right). \tag{3}$$

As a result of Corollary 5, we have that the output $\nu(X) = \nu^{(L)}(X)$ of our SDNN concentrates around its expected value $\mathbb{E}_{(X,Y)\sim\mathscr{D}}[\nu(X)] = \mathbb{E}_{(X,Y)\sim\mathscr{D}}[\nu^{(L)}(X)]$. This proves the first bullet point of Proposition 1.

## 2.2 Martingale inequalities

We assume now that the sequence $\boldsymbol{\nu} = (\nu^{(l)}(X))_{l=0}^{L}$ of outputs of the hidden layers of our SDNN is a martingale; this allows us to state some interesting properties of SDNNs. Note that if $\boldsymbol{\nu}$ is any kind of martingale (very-weak, weak, or strong, see appendix B), then the width of the layers of the SDNN is constant throughout the neural network and equal to $p$. Indeed, if $n_l = n_{l-1} = p$ does not hold for all $l \in \{0, \ldots, L\}$, then computing $\mathbb{E}[\nu^{(l)}(X) \mid \nu^{(0)}(X), \ldots, \nu^{(l-1)}(X)]$ may not have a clear mathematical meaning. Neural networks with constant width are used extensively and a well studied subject (Telgarsky, 2016).

We now show how a martingale hypothesis allows us to weaken the already mild assumption in Theorem 4 and Corollary 5. The price to pay is that the martingale inequalities become looser as the number $l$ of layers increases.[1] This entails that the results we find in this section better suit shallow networks, that is, networks for which $L$ is small. The following theorem is an immediate result of Hayes (2005, Theorem 1.8).[2]

**Theorem 6.** *If $\boldsymbol{\nu}$ is a weak martingale in $\mathbb{R}^p$ such that for every $l \in \{1, \ldots, L\}$, $\|\nu^{(l)}(X) - \nu^{(l-1)}(X)\|_2 \leq M$, for some $M > 0$, then, for every $a > 0$,*

$$P_{(X,Y)\sim\mathscr{D}}\left(\|\nu^{(l)}(X)\|_2 \geq Ma\right) < 2\exp\left(1 - \frac{(Ma-1)^2}{2l}\right). \tag{4}$$

The fact that we demand the distance between the output of successive layers of the neural network to be bounded can be interpreted as the mild requirement for the SDNN to be "smooth". That is, no steep jumps from the output of one layer to the output of the successive one are allowed.

Consider now the sequence $\mathbf{Z} = (Z^{(l)})_{l=0}^{L}$ such that $Z^{(l)} = \nu^{(l)}(X) - \mathbb{E}_{(X,Y)\sim\mathscr{D}}[\nu^{(l)}(X)]$, for all $l \in \{1, \ldots, L\}$. Then, we have the following.

---

[1]This type of inequalities are usually called *tail bounds* in the probability theory literature.
[2]Hayes (2005, Theorem 1.8) is summarized in appendix B.

**Corollary 7.** *If* $\mathbf{Z}$ *is a weak martingale in* $\mathbb{R}^p$ *such that for every* $l \in \{1, \dots, L\}$, $\|Z^{(l)} - Z^{(l-1)}\|_2 \leq M$, *for some* $M > 0$, *then, for every* $a > 0$,

$$
\begin{aligned}
P_{(X,Y)\sim\mathscr{D}}\left(\|Z^{(l)}\|_2 \geq Ma\right) &= P_{(X,Y)\sim\mathscr{D}}\left(\|\nu^{(l)}(X) - \mathbb{E}_{(X,Y)\sim\mathscr{D}}\left[\nu^{(l)}(X)\right]\|_2 \geq Ma\right) \\
&< 2\exp\left(1 - \frac{(Ma-1)^2}{2l}\right).
\end{aligned}
\tag{5}
$$

Requiring the distance between $Z^{(l)}$ and $Z^{(l-1)}$ to be bounded for all $l$ retains the "smoothness" interpretation we have given before: we do not want jumps that are too steep between successive centered outputs of the layers. Corollary 7 tells us that $\nu^{(l)}(X)$ concentrates around $\mathbb{E}_{(X,Y)\sim\mathscr{D}}[\nu^{(l)}(X)]$, for all $l \in \{1, \dots, L\}$. In turn, this provides conditions under which the output of the neural network concentrates around its expected value, thus proving the second bullet point of Proposition 1.

There is one main difference between Corollary 7 and Corollary 5. In the latter we require $\nu^{(l)}(X)$ to be bounded, while in the former we only require that the Euclidean distance between $\nu^{(l)}(X) - \mathbb{E}_{(X,Y)\sim\mathscr{D}}[\nu^{(l)}(X)]$ and $\nu^{(l-1)}(X) - \mathbb{E}_{(X,Y)\sim\mathscr{D}}[\nu^{(l-1)}(X)]$ is bounded. This is a milder assumption since it governs the dynamics of the sequence of layer outputs, rather than that of the outputs themselves. It comes at the cost of requiring that the network is shallow and of verifying that the sequence of outputs in each layer is a weak martingale.

We now provide two theorems that allow us to check whether sequences $\boldsymbol{\nu}$ or $\mathbf{Z}$ are weak or very-weak martingales. We first need a definition.

**Definition 8.** *Consider two generic p-dimensional random vectors* $X_1, X_2$. *If*

$$
\mathbb{E}(\phi(X_1)) \leq \mathbb{E}(\phi(X_2))
$$

*for all convex functions* $\phi : \mathbb{R}^p \to \mathbb{R}$, *provided the expectations exist, then* $X_1$ *is smaller than* $X_2$ *in the convex order, denoted by* $X_1 \leq_{cx} X_2$.

The following results come from Shaked & Shanthikumar (2007, Theorem 7.A.1).

**Theorem 9.** *Pick any* $l \in \{1, \dots, L\}$ *and any* $j < l$. *If* $\nu^{(j)}(X) \leq_{cx} \nu^{(l)}(X)$, *then* $\boldsymbol{\nu}$ *is a weak martingale. Similarly, if* $Z^{(j)} \leq_{cx} Z^{(l)}$, *then* $\mathbf{Z}$ *is a weak martingale.*

**Theorem 10.** *Pick any* $l \in \{1, \dots, L\}$. *If* $\nu^{(l-1)}(X) \leq_{cx} \nu^{(l)}(X)$, *then* $\boldsymbol{\nu}$ *is a very-weak martingale. Similarly, if* $Z^{(l-1)} \leq_{cx} Z^{(l)}$, *then* $\mathbf{Z}$ *is a very-weak martingale.*

To the best of our knowledge, Theorems 9 and 10 are the only existing method for checking whether a vector-valued stochastic processes is a weak or a very-weak martingale. The closest previous procedure derives the distributions of test statistics required for testing the null hypothesis that a given univariate stochastic process is a very-weak martingale (Park & Whang, 2005). To apply it to our case, for every $l \in \{1, \dots, L\}$, we would need to assume that the entries of $\nu^{(l-1)}(X)$ are independent; we would need to assume the same for $Z^{(l-1)}$. This is an extremely strong assumption and unlikely to hold in practice.

**Remark 11.** *For Theorem 6 and Corollary 7 to hold it is enough that* $\boldsymbol{\nu}$ *and* $\mathbf{Z}$ *are a very-weak martingales (Hayes, 2005, Theorem 1.8). However, we require them to be weak martingales for the following result to hold. Pick any* $l \in \{1, \dots, L-1\}$. *Then, we have that*

$$
\begin{aligned}
&\mathbb{E}_{(X,Y)\sim\mathscr{D}}\left[\nu(X) - \mathbb{E}_{(X,Y)\sim\mathscr{D}}(\nu(X)) \mid \nu^{(l)}(X) - \mathbb{E}_{(X,Y)\sim\mathscr{D}}\left(\nu^{(l)}(X)\right)\right] \\
&= \mathbb{E}_{(X,Y)\sim\mathscr{D}}\left[\nu(X) \mid \nu^{(l)}(X) - \mathbb{E}_{(X,Y)\sim\mathscr{D}}\left(\nu^{(l)}(X)\right)\right] - \mathbb{E}_{(X,Y)\sim\mathscr{D}}\left[\nu(X)\right] \\
&= \nu^{(l)}(X) - \mathbb{E}_{(X,Y)\sim\mathscr{D}}\left[\nu^{(l)}(X)\right],
\end{aligned}
\tag{6}
$$

*where the last equality comes from the weak martingale property of* $\mathbf{Z}$. *Then, the equalities in equation 6 imply that, for all* $a > 0$,

$$
\begin{aligned}
&P_{(X,Y)\sim\mathscr{D}}\left(\|\mathbb{E}_{(X,Y)\sim\mathscr{D}}\left[\nu(X) \mid \nu^{(l)}(X) - \mathbb{E}_{(X,Y)\sim\mathscr{D}}\left(\nu^{(l)}(X)\right)\right] - \mathbb{E}_{(X,Y)\sim\mathscr{D}}[\nu(X)]\|_2 \geq Ma\right) \\
&= P_{(X,Y)\sim\mathscr{D}}\left(\|\nu^{(l)}(X) - \mathbb{E}_{(X,Y)\sim\mathscr{D}}\left[\nu^{(l)}(X)\right]\|_2 \geq Ma\right) < 2\exp\left(1 - \frac{(Ma-1)^2}{2l}\right),
\end{aligned}
\tag{7}
$$

*where the inequality comes from Corollary 7. Equation equation 7 tells us that we can approximate the value* $\mathbb{E}_{(X,Y)\sim\mathscr{D}}[\nu(X)]$ *that the output* $\nu(X)$ *of our SDNN concentrates around with the quantity* $\mathbb{E}_{(X,Y)\sim\mathscr{D}}[\nu(X) \mid \nu^{(l)}(X) - \mathbb{E}_{(X,Y)\sim\mathscr{D}}(\nu^{(l)}(X))]$. *The approximation is good for the first layers, and then becomes coarse, since as the number l of layers increases, so does the bound. This estimate, then, is better in the context of shallow networks.*

## 2.3 Classification accuracy

In this section, we focus on binary classification for the sake of exposition. As we pointed out earlier, a DNN $\nu : \mathbb{R}^d \to \mathbb{R}^p$, together with a choice of score function $s : \mathbb{R}^p \to \mathbb{R}$, gives us a classifier. If the score function $s$ computed at the output value $\nu(x)$, $s(\nu(x))$, exceeds some decision threshold $c$, then the neural network predicts $x$ is from a certain class $\mathscr{C}_1$, otherwise $x$ is from the other category $\mathscr{C}_2$. The input space is then partitioned into two disjoint subsets by the decision boundary $\mathcal{B} := \{x \in \mathbb{R}^d : \nu(x) = s^{-1}(c)\}$. We call connected regions with value above the threshold *positive regions*, while those having value below the threshold *negative regions*.

For the sake of exposition suppose that $n_L = 1$, that is, $\nu : \mathbb{R}^d \to \mathbb{R}$. Then, in our stochastic setting, we have that $s(\nu(X))$ is a random variable that maps into $\mathbb{R}$; we assume it has finite first moment. We have the following important result.

**Proposition 12.** *If* $a \leq s(\nu(X)) \leq b$ *with probability* 1*, for some* $a, b \in \mathbb{R}$, $a < b$, *and* $c \in [a, b]$, *then for all* $t > 0$,

$$P_{(X,Y)\sim\mathscr{D}}\left(\left|s(\nu(X)) - \mathbb{E}_{(X,Y)\sim\mathscr{D}}[s(\nu(X))]\right| \geq t\right) \leq 2\exp\left(-\frac{2t^2}{(b-a)^2}\right).$$

Proposition 12 tells us that $s(\nu(X))$ concentrates around its expected value. The fact that $s(\nu(X))$ is bounded is a mild condition, as long as $c \in [a, b]$; it simply amounts to the choice of the score function.

We now consider the *expected classifier* based on the *expected decision boundary* (EDB)

$$\mathcal{B}_{exp} := \left\{x \in \mathbb{R}^d : \mathbb{E}_{(X,Y)\sim\mathscr{D}}[s(\nu(X))] = c\right\},$$

for some decision threshold $c$. That is,

(i) if $\mathbb{E}_{(X,Y)\sim\mathscr{D}}[s(\nu(X))] > c$, then $x \in \mathscr{C}_1$;

(ii) if $\mathbb{E}_{(X,Y)\sim\mathscr{D}}[s(\nu(X))] < c$, then $x \in \mathscr{C}_2$.

Note that being based on the value $\mathbb{E}_{(X,Y)\sim\mathscr{D}}[s(\nu(X))]$, the expected classifier will most likely not be perfect. We can provide a probabilistic bound for the classification error of the expected classifier.

**Proposition 13.** *If* $\mathbb{E}_{(X,Y)\sim\mathscr{D}}[s(\nu(X))] > c$, *then there exists* $t_1 > 0$ *such that*

$$P_{(X,Y)\sim\mathscr{D}}\left(s(\nu(X)) \leq c\right) \leq \exp\left(-\frac{2{t_1}^2}{(b-a)^2}\right). \tag{8}$$

*If instead* $\mathbb{E}_{(X,Y)\sim\mathscr{D}}[s(\nu(X))] < c$, *then there exists* $t_2 > 0$, *possibly different than* $t_1$, *such that*

$$P_{(X,Y)\sim\mathscr{D}}\left(s(\nu(X)) \geq c\right) \leq \exp\left(-\frac{2{t_2}^2}{(b-a)^2}\right). \tag{9}$$

Call $p_1$ the bound in equation 8 and $p_2$ the one in equation 9. Then, Proposition 13 tells us that the expected classifier is correct with probability $p > 1 - \max\{p_1, p_2\}$; together with Proposition 12, it proves Proposition 2.

## 3 Optimal number of layers

We find the number of layers for a generic SDNN that strikes the perfect balance between accuracy of the analysis and computational cost. We do so using an optimal stopping technique. We assume that $\nu$ has constant width $p$ for the same reasons as in section 2.2.

Sequence $\boldsymbol{\nu} = (\nu^{(l)}(X))_{l=0}^{L}$ induces a stochastic process $\boldsymbol{\gamma} = (\gamma^{(l)})_{l=0}^{L}$, where for every $l$, $\gamma^{(l)}$ is a real-valued random variable, $\gamma^{(l)} : \Omega \to \mathbb{R}$. We interpret $\gamma^{(l)}$ as the utility of choosing $l$ many layers for the neural network, that is, of stopping the observation of $\boldsymbol{\gamma}$ at layer $l$. For example, consider the following situation. Assume that, for all $l_1, l_2 \in \{1, \dots, L\}$,

$$l_1 \leq l_2 \implies \|\nu^{(l_1)}(X) - Y^\star\|_2 \geq \|\nu^{(l_2)}(X) - Y^\star\|_2, \tag{10}$$

where $Y^\star$ is the correct response to input $X$. Equation equation 10 means that the deeper the neural network is, the closer its output is to the truth. This assumption reflects the "accuracy of the analysis" side of the problem.

Now, write the loss function used in the study as a function of the number of layers. So for example the mean squared error (MSE) loss is written as

$$\text{LOSS}(l) = \frac{1}{p}\|\nu^{(l)}(X) - Y^\star\|_2^2. \tag{11}$$

Suppose then that for our analysis we choose a loss function that is positive and monotonically decreasing in $l$. The importance of the monotonicity assumption is discussed in a few lines. Given our assumption equation 10, the MSE loss in equation 11 is a sensible choice.

Let $\mathfrak{g}$ be a generic functional on $\mathbb{R}$ such that $\mathfrak{g} \circ \text{LOSS}$ is a positive monotonically decreasing function on $l$. In our example, we can let $\mathfrak{g}$ be the identity function.

Finally, consider a generic functional $\mathfrak{h}$ on $\{1, \dots, L\}$ that is positive and monotonically decreasing in $l$. In our example, we can take $\mathfrak{h}(l) := \frac{1}{c\sqrt{l}}$, for some $c > 0$.

Then, we let $\gamma^{(0)} = 0$, and for all $l \in \{1, \dots, L\}$,

$$\gamma^{(l)} = (\mathfrak{g} \circ \text{LOSS})(l) \cdot \mathfrak{h}(l).$$

In our example, then, $\gamma^{(l)} = \text{LOSS}(l)/[c\sqrt{l}]$. This is a reasonable choice for $\gamma^{(l)}$ because the loss is smaller as $l$ increases, but at the same time values of $l$ that are too large are penalized. The optimal number of layers gives us the balance between cost and accuracy that we are looking for.

In this example, for every $l \in \{1, \dots, L\}$, $\gamma^{(l)}$ is the product of two positive monotonically decreasing functions in $l$ defined on the finite domain $\{1, \dots, L\}$. This means that $\gamma^{(l)}$ itself is positive and monotonically decreasing in $l$. This is important from an application point of view: if $\gamma^{(l)}$ were not to be monotonic, then all possible values of the number of layers would have to be explored to find the optimum. The choice of functions $\mathfrak{g}$ and $\mathfrak{h}$ gives the scholar control on the trade-off: if more accuracy is required, then $\mathfrak{g}$ will be chosen so that $\mathfrak{g} \circ \text{LOSS}$ decreases faster than $\mathfrak{h}$, and vice versa if the computational aspect is more important.

Then, we consider the natural filtration $\mathbf{F}^\gamma$ of $\mathcal{F}$ with respect to $\boldsymbol{\gamma}$ given by

$$\mathcal{F}_l^\gamma := \sigma\left(\left\{\left(\gamma^{(s)}\right)^{-1}(A) : s \in \mathbb{N}_0,\, s \leq l,\, A \in \mathcal{B}(\mathbb{R})\right\}\right),$$

where $\mathbb{N}_0 := \mathbb{N} \cup \{0\}$.[3] We assume that $\boldsymbol{\gamma}$ is adapted to filtration $\mathbf{F}^\gamma$, that is, $\gamma^{(l)}$ is $\mathcal{F}_l^\gamma$-measurable, for all $l$. We interpret $\mathcal{F}_l^\gamma$ as the information available up to layer $l$. Our decision regarding whether to choose $l$ many layers – that is, to stop observing $\boldsymbol{\gamma}$ at layer $l$ – must be based on this information only (no anticipation is allowed).

---

[3]For the definitions of filtration and natural filtration, see appendix B.

**Definition 14.** *Given a generic filtered probability space $(\Omega, \mathcal{F}, (\mathcal{F}_l)_{l \in \mathbb{N}_0}, P)$, a random variable $\tau : \Omega \to \mathbb{N}_0 \cup \{\infty\}$ is called a Markov time if $\{\tau \leq l\} \in \mathcal{F}_l$, for all $l \in \mathbb{N}_0$. A Markov time is called a stopping time if $\tau < \infty$ P-a.s.*

We denote the family of all stopping times by $\mathfrak{M}$, and the family of all Markov times by $\overline{\mathfrak{M}}$. A family that we are going to use later in this section is $\mathfrak{M}_l^L := \{\tau \in \mathfrak{M} : l \leq \tau \leq L\}$, $0 \leq l \leq L$. For notational convenience, we let $\mathfrak{M}^L \equiv \mathfrak{M}_0^L$.

The optimal stopping problem we study is the following

$$V_\star = \sup_\tau \mathbb{E}_{(X,Y) \sim \mathscr{D}} \left[ \gamma^{(\tau)} \right] \tag{12}$$

where the supremum is taken over a family of stopping times. Note that equation 12 involves two tasks, that is computing the value function $V_\star$ as explicitly as possible, and finding an optimal stopping time $\tau_\star$ at which the supremum is attained.

To ensure that the expected value in equation 12 exists, we need a further assumption, that is

$$\mathbb{E}_{(X,Y) \sim \mathscr{D}} \left[ \sup_{l \leq k \leq L} \left| \gamma^{(k)} \right| \right] < \infty, \tag{13}$$

If equation 13 is satisfied, then $\mathbb{E}_{(X,Y) \sim \mathscr{D}}[\gamma^{(\tau)}(x)]$ is well defined for all $\tau \in \mathfrak{M}_l^L$.

To each of the families $\mathfrak{M}_l^L$ we assign the following value function

$$V_l^L := \sup_{\tau \in \mathfrak{M}_l^L} \mathbb{E}_{(X,Y) \sim \mathscr{D}} \left[ \gamma^{(\tau)} \right], \tag{14}$$

$0 \leq l \leq L$. For notational convenience, we let $V^L \equiv V_0^L$.

We solve problem equation 14 using the backward induction approach outlined in Peskir & Shiryaev (2006, Section 1.2). We assume $L < \infty$, but this is without loss of generality. Notice that equation 14 can be rewritten as

$$V_l^L = \sup_{l \leq \tau \leq L} \mathbb{E}_{(X,Y) \sim \mathscr{D}} \left[ \gamma^{(\tau)} \right], \tag{15}$$

where $\tau$ is a stopping time and $0 \leq l \leq L$. We solve the problem by letting time go backwards; we proceed recursively. Let $L$ be a high number in $\mathbb{N}$, e.g. 1000. It is going to represent the maximum number of layers we deem "usable" for our neural network. We consider an ancillary sequence of random variables $(S_l^L)_{0 \leq l \leq L}$ induced by $\gamma$ that is built as follows. For $l = L$ we stop, and our utility is $S_l^L = \gamma^{(L)}$; for $l = L - 1$, we can either stop or continue. If we stop, our utility is $S_{L-1}^L = \gamma^{(L-1)}$, while if we continue our utility is $S_{L-1}^L = \mathbb{E}_{(X,Y) \sim \mathscr{D}}[S_L^L \mid \mathcal{F}_{L-1}^\gamma]$. As it is clear from the latter conclusion, our decision about stopping at layer $l = L - 1$ or continuing with an extra layer must be based on the information contained in $\mathcal{F}_{L-1}^\gamma$ only. So, if $\gamma^{(L-1)} \geq \mathbb{E}_{(X,Y) \sim \mathscr{D}}[S_L^L \mid \mathcal{F}_{L-1}^\gamma]$, we stop at layer $L-1$, otherwise we add an extra layer. For $l \in \{L-2, \ldots, 0\}$ the considerations are continued analogously.

By the backward induction method we just described, we have that the elements of the sequence $(S_l^L)_{0 \leq l \leq L}$ are defined recursively as

$$S_l^L = \gamma^{(L)}, \quad \text{for } l = L, \tag{16}$$

$$S_l^L = \max \left\{ \gamma^{(l)}, \mathbb{E}_{(X,Y) \sim \mathscr{D}} \left[ S_{l+1}^L \mid \mathcal{F}_l^\gamma \right] \right\}, \quad \text{for } l \in \{L-1, \ldots, 0\}. \tag{17}$$

The method also suggests that we consider the following stopping time

$$\tau_l^L = \inf_{l \leq k \leq L} \left\{ S_k^L = \gamma^{(k)} \right\}, \tag{18}$$

for all $0 \leq l \leq L$. Notice that the infimum is always attained. The following is the main result of this section; it tells us that $\tau_l^L$ is indeed the optimal stopping time for problem equation 15. It comes from Peskir & Shiryaev (2006, Theorem 1.2).

**Theorem 15.** *Consider the optimal stopping problem equation 15, and assume that equation 13 holds. Then, for all $l \in \{0, \ldots, L\}$, we have that*

$$S_l^L \geq \mathbb{E}_{(X,Y)\sim\mathscr{D}}\left[\gamma^{(\tau)} \mid \mathcal{F}_l^\gamma\right], \quad \text{for all } \tau \in \mathfrak{M}_l^L, \tag{19}$$

$$S_l^L = \mathbb{E}_{(X,Y)\sim\mathscr{D}}\left[\gamma^{\left(\tau_l^L\right)} \mid \mathcal{F}_l^\gamma\right]. \tag{20}$$

*In addition, fix any $l \in \{0, \ldots, L\}$. Then,*

    *(i) the stopping time $\tau_l^L$ is optimal for equation 15;*

    *(ii) if $\tau_\star$ is any optimal stopping time for equation 15, then $\tau_l^L \leq \tau_\star \ P_{(X,Y)\sim\mathscr{D}}$-a.s.;*

    *(iii) The stopped sequence $(S_{k \wedge \tau_l^L}^L)_{l \leq k \leq L}$ is a strong martingale.*

It follows immediately from Theorem 15.(i) that the optimal number of layers for our neural network $\nu$ is given by $\tau_1^L$. This proves Proposition 3.

## 4 Application: stochastic feedforward neural networks with ReLU activations

In this section we apply our results for a general SDNN to a stochastic version of the feedforward neural network with ReLU activation (FNNRA) introduced in Zhang et al. (2018). The key insight in Zhang et al. (2018) is that ideas from tropical algebra can be used to study feedforward neural networks, especially with ReLU activations. The intuition is that the activation function of a feedforward neural network requires computing a maximum, which turns out to correspond to tropical addition.

We first present the notation we use for feedforward neural networks. A short summary of ideas from tropical algebra is given in appendix C.

### 4.1 Deterministic feedforward neural networks

We use this section to fix the notation for feedforward neural networks. We then introduce the the FNNRA proposed in Zhang et al. (2018). We restrict our attention to fully connected feedforward neural networks.

An $L$-layered feedforward neural network is a map $\nu : \mathbb{R}^d \to \mathbb{R}^p$ given by a composition of functions

$$\nu \equiv \nu^{(L)} = \sigma^{(L)} \circ \rho^{(L)} \circ \sigma^{(L-1)} \circ \rho^{(L-1)} \circ \cdots \circ \sigma^{(1)} \circ \rho^{(1)}.$$

The *preactivation functions* $\rho^{(1)}, \ldots, \rho^{(L)}$ are affine transformations to be determined, and the *activation functions* $\sigma^{(1)}, \ldots, \sigma^{(L)}$ are chosen and fixed in advance. Affine function $\rho^{(l)} : \mathbb{R}^{n_{l-1}} \to \mathbb{R}^{n_l}$ is given by a weight matrix $A^{(l)} \in \mathbb{Z}^{n_l \times n_{l-1}}$ and a bias vector $b^{(l)} \in \mathbb{R}^{n_l}$,

$$\rho^{(l)}(\nu^{(l-1)}) := A^{(l)}\nu^{(l-1)} + b^{(l)}.$$

The $(i, j)$-th coordinate of $A^{(l)}$ is denoted by $a_{ij}^{(l)}$, and the $i$-th coordinate of $b^{(l)}$ by $b_i^{(l)}$. Collectively they form the *parameters* of the $l$-th layer. Notice that, for all $l$, $A^{(l)}$ can be decomposed as a difference of two nonnegative integer valued matrices, $A^{(l)} = A_+^{(l)} - A_-^{(l)}$, with $A_+^{(l)}, A_-^{(l)} \in \mathbb{Z}^{n_l \times n_{l-1}}$, so that their entries are

$$a_{+ij}^{(l)} = \max\{a_{ij}, 0\}, \quad a_{-ij}^{(l)} = \max\{-a_{ij}, 0\}.$$

For a vector input $x \in \mathbb{R}^{n_l}$, $\sigma^{(l)}(x)$ is understood to be in coordinatewise sense. We make the following assumptions on the architecture of our feedforward neural network:

    (a) the weight matrices $A^{(1)}, \ldots, A^{(L)}$ are integer-valued;

    (b) the bias vectors $b^{(1)}, \ldots, b^{(L)}$ are real-valued;

(c) the activation functions $\sigma^{(1)}, \ldots, \sigma^{(L)}$ take the form

$$\sigma^{(l)}(x) := \max\{x, t^{(l)}\} = \left( \max\{x_1, t_1^{(l)}\}, \ldots, \max\{x_{n_l}, t_{n_l}^{(l)}\} \right)^\top,$$

where $t^{(l)} \in (\mathbb{R} \cup \{-\infty\})^{n_l}$ is the *threshold vector*, and $x_j$ and $t_j^{(l)}$ denote the $j$-th element of $x$ and $t^{(l)}$, respectively.

We assume all the neural networks in this section to satisfy (a)-(c).

As pointed out in Zhang et al. (2018, Section 4), assumption (b) is general, and yields no loss of generality. The same goes for (a), since:

- real weights can be approximated arbitrarily closely by rational weights;

- one may "clear denominators" in these rational weights by multiplying them by the least common multiple of their denominators to obtain integer weights;

- scaling all weights and biases by the same positive constant does not influence the workings of a neural network.

The form of the activation function in (c) includes both ReLU activation ($t^{(l)} = 0\mathbf{1}_{n_l}$) and identity map ($t^{(l)} = -\infty\mathbf{1}_{n_l}$, so that $\sigma^{(l)}(x) = x$) as special cases, where $\mathbf{1}_{n_l}$ is vector $(1, \ldots, 1)^\top$ having $n_l$ entries. We only consider the ReLU activation function in this section as we are generalizing the model in Zhang et al. (2018) where the tropical algebra makes the most sense for ReLU activations. In addition, much of the theory literature on neural networks has focused on ReLU networks (Arora et al., 2018; Montúfar et al., 2014; Zhang et al., 2018).

Let us now describe the the deterministic ReLU neural network proposed in Zhang et al. (2018); its architecture is depicted in Figure 1.[4] In Zhang et al. (2018, Section 5), the authors state that the $(l+1)$-th layer $\nu^{(l+1)}(x)$ of an $L$-layered FNNRA $\nu(x)$ can be written as $\nu^{(l+1)}(x) = F^{(l+1)}(x) - G^{(l+1)}(x)$, for all $x \in \mathbb{R}^d$, where

$$
\begin{aligned}
F^{(l+1)}(x) &= H^{(l+1)}(x) \oplus G^{(l+1)}(x) \odot t^{(l+1)} = \max\{H^{(l+1)}(x), G^{(l+1)}(x) + t^{(l+1)}\}, \\
G^{(l+1)}(x) &= A_+^{(l+1)} G^{(l)}(x) \odot A_-^{(l+1)} F^{(l)}(x) = A_+^{(l+1)} G^{(l)}(x) + A_-^{(l+1)} F^{(l)}(x), \\
H^{(l+1)}(x) &= A_+^{(l+1)} F^{(l)}(x) \odot A_-^{(l+1)} G^{(l)}(x) \odot b^{(l+1)} = A_+^{(l+1)} F^{(l)}(x) + A_-^{(l+1)} G^{(l)}(x) + b^{(l+1)},
\end{aligned}
\tag{21}
$$

where $F^{(l)}(x)$ and $G^{(l)}(x)$ are vectors in $\mathbb{R}^{n_l}$ whose coordinates are tropical polynomials in $x$.[5] That is,

$$
\begin{aligned}
F^{(l)}(x) &= \left( f_1^{(l)}(x), \ldots, f_{n_l}^{(l)}(x) \right)^\top, \\
G^{(l)}(x) &= \left( g_1^{(l)}(x), \ldots, g_{n_l}^{(l)}(x) \right)^\top,
\end{aligned}
\tag{22}
$$

where

$$
\begin{aligned}
f_j^{(l)}(x) &= \max\{c_{j1} x^{\odot \alpha_{j1}}, \ldots, c_{jr} x^{\odot \alpha_{jr}}\}, \\
g_j^{(l)}(x) &= \max\{c'_{j1} x^{\odot \alpha'_{j1}}, \ldots, c'_{jr} x^{\odot \alpha'_{jr}}\},
\end{aligned}
\tag{23}
$$

---

[4]Figure 1 is a replica of Zhang et al. (2018, Figure A.1).

[5]Notice that we can write the $i$-th entry of vector $A_+^{(l+1)} G^{(l)}(x)$ in tropical notation as

$$\bigodot_{j \in \{1, \ldots, n_l\}} \left[ a_{+ij}^{(l+1)} \right]^{\odot g_j^{(l)}(x)}.$$

We can write similarly the $i$-th entries of $A_-^{(l+1)} G^{(l)}(x)$, $A_+^{(l+1)} F^{(l)}(x)$, and $A_-^{(l+1)} F^{(l)}(x)$.

for all $j \in \{1, \ldots, n_l\}$ and some $r \in \mathbb{N}$, and

$$c_{js}x^{\odot \alpha_{js}} := c_{js} + \alpha_{js,1}x_1 + \alpha_{js,2}x_2 + \ldots + \alpha_{js,d}x_d,$$

for all $j \in \{1, \ldots, n_l\}$ and all $s \in \{1, \ldots, r\}$. Similarly,

$$c'_{js}x^{\odot \alpha'_{js}} := c'_{js} + \alpha'_{js,1}x_1 + \alpha'_{js,2}x_2 + \ldots + \alpha'_{js,d}x_d,$$

for all $j \in \{1, \ldots, n_l\}$ and all $s \in \{1, \ldots, r\}$. Notice that $c_{js}, c'_{js} \in \mathbb{R}$ for all $j$ and all $s$, and that $\alpha_{js} = (\alpha_{js,1}, \ldots, \alpha_{js,d})^\top, \alpha'_{js} = (\alpha'_{js,1}, \ldots, \alpha'_{js,d})^\top \in \mathbb{N}^d$, for all $j$ and all $s$.

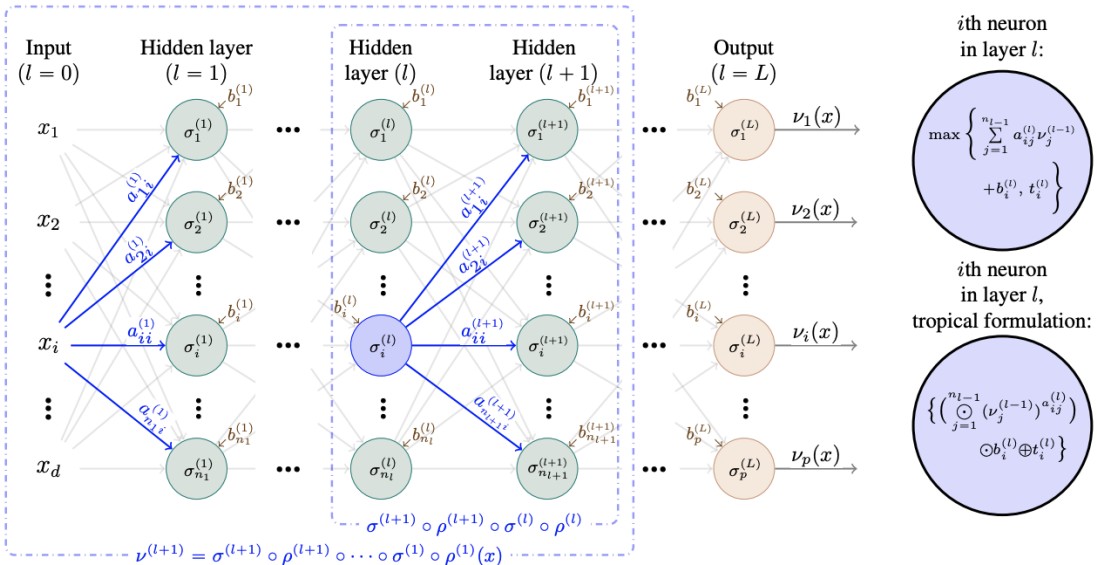

Figure 1: Architecture of the FNNRA $\nu : \mathbb{R}^d \to \mathbb{R}^p$ with $L$ layers proposed in Zhang et al. (2018).

## 4.2 Stochastic feedforward ReLU networks

In this section we derive a stochastic version of the FNNRA described in section 4.1. We introduce two sources of of stochasticity: (1) the initialization or the starting values of the neural network are random (that is, $F^{(0)}(X)$ and $G^{(0)}(X)$ are random vectors), and (2) the parameters of all the layers are aleatory (that is, $A^{(l)}$ is a random matrix with integer entries, and $b^{(l)}$ is a random vector with real entries, for all $l \in \{1, \ldots, L\}$). We now state the update rules for the first layer, and the second layer given the first. By induction, this is enough to specify the stochastic feedforward network.

For the first layer we sample random vectors $F^{(0)}(X)$ and $G^{(0)}(X)$ from a distribution $\mathscr{R}$ on $\mathbb{R}^{n_0}$ that is built as follows. Sample

$$(c_{j1}, \ldots, c_{jr})^\top, (c'_{j1}, \ldots, c'_{jr})^\top \sim \mathscr{S}_j \text{ on } \mathbb{R}^r, \quad \alpha_{j1}, \ldots, \alpha_{jr}, \alpha'_{j1}, \ldots, \alpha'_{jr} \sim \mathscr{T}_j \text{ on } \mathbb{N}^d,$$

for $j \in \{1, \ldots, n_0\}$. Note that we do not require the $\alpha_{js}$'s and the $\alpha'_{js}$'s to be iid and we do not require the samples to be independent across the index $j$. Then, $F^{(0)}(x)$ and $G^{(0)}(x)$ – the realizations of $F^{(0)}(X)$ and $G^{(0)}(X)$, respectively – are computed according to equation 22 and equation 23. Notice that $F^{(0)}(X)$ and $G^{(0)}(X)$ need not be independent.

To compute the subsequent layer we first sample $A^{(l+1)} \sim \mathscr{P}$ on $\mathbb{Z}^{n_{l+1} \times n_l}$ and $b^{(l+1)} \sim \mathscr{Q}$ on $\mathbb{R}^{n_{l+1}}$, for $l \in \{0, \ldots, L-1\}$. After computing $F^{(l+1)}(x)$ and $G^{(l+1)}(x)$ from $F^{(l)}(x)$ and $G^{(l)}(x)$ by equation equation 21, we find $\nu^{(l+1)}(x) = F^{(l)}(x) - G^{(l)}(x)$. Note that $\nu^{(l+1)}(x)$ is the realization of random vector $\nu^{(l+1)}(X) : \Omega \to \mathbb{R}^{n_{l+1}}$ whose elements may be correlated and that we assume to have finite fist moment. We now apply Corollary 5 to our stochastic FNNRA.

**Proposition 16.** *If $\mathscr{P}$, $\mathscr{Q}$, $\mathscr{S}_j$, and $\mathscr{T}_j$, $j \in \{1, \ldots, n_0\}$ are bounded, then for all $l \in \{1, \ldots, L\}$, there exists $\xi_{(l)} \in \mathbb{R}$ such that $\|\nu^{(l)}(X)\|_2 \leq \xi_{(l)}$ and equation 3 holds.*

The assumption that $\mathscr{P}$, $\mathscr{Q}$, $\mathscr{S}_j$, and $\mathscr{T}_j$, $j \in \{1, \ldots, n_0\}$, are bounded should always be verified: even if the distributions we want to use are unbounded, we can always truncate them and use the truncated versions to obtain the concentration result in Proposition 16.

**Remark 17.** *Notice that if $t^{(l)}$ were a random quantity distributed according to $\mathscr{V}$ on $\mathbb{R}^{n_l}$, and possibly correlated with other $t^{(k)}$'s, Proposition 16 would still hold, provided that $\mathscr{V}$ is bounded.*[6]

### 4.3 Concentration inequality for positive and negative regions

Tropical rational functions are piecewise linear, so the notion of linear regions applies. A *linear region* of a tropical rational function $f$ is a maximal connected subset of the domain on which $f$ is linear; the number of linear regions of $f$ is denoted by $\mathcal{N}(f)$.

In Zhang et al. (2018, Corollary 5.3, Theorem 5.4, and Proposition 5.5), the authors show the equivalence of tropical rational functions, continuous piecewise linear functions with integer coefficients, and neural networks satisfying assumptions (a)-(c). Hence, they are able to link the number of linear regions of a tropical rational function to (bounds on) the number of positive and negative regions that the neural network divides the input space into.[7]

**Proposition 18.** *(Zhang et al., 2018, Proposition 6.1) Let $\nu$ be an L-layer stochastic neural network satisfying (a)-(c) in section 4.1 such that $t^{(L)} = -\infty$ and $p = n_L = 1$, that is, $\nu : \mathbb{R}^d \to \mathbb{R}$. Let the score function $s : \mathbb{R} \to \mathbb{R}$ be injective with decision threshold $c$ in its range. If $\nu = f \oslash g$, where $f$ and $g$ are tropical polynomials, then the number of connected positive regions is at most $\mathcal{N}(f)$, while the number of connected negative regions is at most $\mathcal{N}(g)$.*

Suppose now all the assumptions of Proposition 18 hold. In our stochastic setting, we have that $f$ and $g$ are both random variables, so $\mathcal{N}(f)$ and $\mathcal{N}(g)$ are two stochastic quantities in $\mathbb{N}$. Assume their first moment is finite and notice that they are bounded below by 1. Then, the following proposition holds.

**Proposition 19.** *If $\mathcal{N}(f) \leq b_1$ and $\mathcal{N}(g) \leq b_2$ with probability 1, for some possibly different natural numbers $b_1, b_2 > 1$, then for all $t > 0$,*

$$P_{(X,Y)\sim\mathscr{D}} \left( \left| \mathcal{N}(f) - \mathbb{E}_{(X,Y)\sim\mathscr{D}}[\mathcal{N}(f)] \right| \geq t \right) \leq 2 \exp\left( -\frac{2t^2}{(b_1 - 1)^2} \right)$$

*and*

$$P_{(X,Y)\sim\mathscr{D}} \left( \left| \mathcal{N}(g) - \mathbb{E}_{(X,Y)\sim\mathscr{D}}[\mathcal{N}(g)] \right| \geq t \right) \leq 2 \exp\left( -\frac{2t^2}{(b_2 - 1)^2} \right).$$

The assumption that $\mathcal{N}(f)$ and $\mathcal{N}(g)$ have an upper bound is always verified: all the possible realizations of $f$ and $g$ have a finite number of linear regions. Proposition 19 tells us that the upper bounds for the number of positive and negative regions concentrate around their expected value.

## 5 Conclusion

In this paper we present concentration inequalities for the hidden layers and the output of a generic SDNN based on the ideas of norm-sub-Gaussian distribution and of weak martingale. We introduce the notion of expected classifier and give a probabilistic bound to its classification error. We also find – via an optimal stopping procedure – the number of layers for the SDNN that strikes the perfect balance between computational cost and accuracy of the analysis. Finally, we apply our findings to a stochastic version of the FNNRA of Zhang et al. (2018). In future work, we plan to explore the geometric properties of SDNNs, in the spirit of Alfarra et al. (2021); Hauser & Ray (2017); Maragos et al. (2021).

---

[6]If $t^{(l)} \sim \mathscr{V}$ on $\mathbb{R}^{n_l}$, this means that we rule out the identity activation map. To include it, $\mathscr{V}$ should be a distribution on $(\mathbb{R} \cup \{-\infty\})^{n_l}$.

[7]Recall that positive and negative regions were introduced in section 2.3.

## Acknowledgements

We would like to thank Federico Ferrari, Vittorio Orlandi, Alessandro Zito, and three anonymous referees for their helpful comments. Michele Caprio would like to acknowledge partial funding from NSF CCF-1934964 and ARO MURI W911NF2010080. Sayan Mukherjee would like to acknowledge partial funding from HFSP RGP005, NSF DMS 17-13012, NSF BCS 1552848, NSF DBI 1661386, NSF IIS 15-46331, NSF DMS 16-13261, and the Alexander von Humboldt Foundation. High-performance computing is partially supported by grant 2016-IDG-1013 from the North Carolina Biotechnology Center. Sayan Mukherjee would also like to acknowledge the German Federal Ministry of Education and Research within the project Competence Center for Scalable Data Analytics and Artificial Intelligence (ScaDS.AI) Dresden/Leipzig (BMBF 01IS18026B).

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

# A   Sub-Gaussian random variables and norm-sub-Gaussian random vectors

Sub-Gaussian random variables and norm-sub-Gaussian random vectors are crucial concepts for the results in section 2.1. We first introduce the former (Jin et al., 2019).

**Definition 20.** *A generic random variable $X \in \mathbb{R}$ is sub-Gaussian, written $SG(\sigma)$, if there exists $\sigma \in \mathbb{R}$ such that, for all $\theta \in \mathbb{R}$,*

$$\mathbb{E}\left[e^{\theta(X-\mathbb{E}(X))}\right] \leq \exp\left(\frac{\theta^2\sigma^2}{2}\right).$$

The following is an important characterization of sub-Gaussian random variables.

**Proposition 21.** *If $X$ is $SG(\sigma)$, then for all $t > 0$*

$$P\left(|X - \mathbb{E}(X)| \geq t\right) \leq 2\exp\left(-\frac{t^2}{2\sigma^2}\right).$$

If a random variable is bounded, then it is sub-Gaussian.

**Proposition 22.** *Let $X$ be a generic random variable in $\mathbb{R}$ such that $a \leq X \leq b$ with probability $1$, for some $a, b \in \mathbb{R}$, $a < b$. Then, for all $\theta \in \mathbb{R}$,*

$$\mathbb{E}\left[e^{\theta(X - \mathbb{E}(X))}\right] \leq \exp\left(\frac{\theta^2}{2}\frac{(b-a)^2}{4}\right),$$

*that is, $X$ is $SG\left(\frac{b-a}{2}\right)$.*

We now give the random vector counterpart of a sub-Gaussian random variable.

**Definition 23.** *A generic random vector $X \in \mathbb{R}^d$ is norm-sub-Gaussian, written $nSG(\sigma)$, if there exists $\sigma \in \mathbb{R}$ such that, for all $t \in \mathbb{R}$,*

$$P\left(\|X - \mathbb{E}(X)\|_2 \geq t\right) \leq 2\exp\left(-\frac{t^2}{2\sigma^2}\right).$$

Definition 23 is closely related to the characterization in Proposition 21. The following comes from Jin et al. (2019, Lemma 1), and it is the random vector counterpart of Proposition 22.

**Proposition 24.** *Let $X$ be a generic bounded random vector in $\mathbb{R}^d$ such that $\|X\|_2 \leq \sigma$. Then, $X$ is $nSG(\sigma)$.*

## B  Martingales and filtrations

We first define very-weak, weak, and strong martingales. They were introduced in Hayes (2005), and are extremely important for section 2.2. We denote by $\mathbf{E}$ any real Euclidean space (of finite or infinite dimension), and by $\mathbf{1_E}$ the vector $(1, 1, \ldots, 1)^\top$ having $\#\mathbf{E}$ entries, where $\#$ denotes the cardinality operator.

**Definition 25.** *Let $\mathbf{X} = (X_j : \Omega \to \mathbf{E})$ be a sequence of random vectors taking values in $\mathbf{E}$, such that $X_0 = 0\mathbf{1_E}$, and for every $i \geq 1$, $\mathbb{E}(\|X_j\|_2) < \infty$ and $\mathbb{E}(X_j \mid X_0, \ldots, X_{j-1}) = X_{j-1}$. Then we call $\mathbf{X}$ a strong martingale in $\mathbf{E}$.*

**Definition 26.** *Let $\mathbf{X} = (X_j : \Omega \to \mathbf{E})$ be a sequence of random vectors taking values in $\mathbf{E}$, such that $X_0 = 0\mathbf{1_E}$, and for every $j < i$, $\mathbb{E}(\|X_j\|_2) < \infty$ and $\mathbb{E}(X_j \mid X_j) = X_j$. Then we call $\mathbf{X}$ a weak martingale in $\mathbf{E}$.*

If $\mathbf{X}$ is a strong martingale, then it is also a weak martingale. The converse need not hold.

**Definition 27.** *Let $\mathbf{X} = (X_j : \Omega \to \mathbf{E})$ be a sequence of random vectors taking values in $\mathbf{E}$, such that $X_0 = 0\mathbf{1_E}$, and for every $i \geq 1$, $\mathbb{E}(\|X_j\|_2) < \infty$ and $\mathbb{E}(X_j \mid X_{j-1}) = X_{j-1}$. Then we call $\mathbf{X}$ a very-weak martingale in $\mathbf{E}$.*

If $\mathbf{X}$ is a weak martingale, then it is also a very-weak martingale. The converse need not hold. Notice that in section 2.2 we implicitly assume $\nu^{(0)}(X) = 0\mathbf{1}_p$. This is just a convention, even if it differs from the one in section 2.1 where we require $\nu^{(0)}(X)$ to be equal to $X$. We now state Hayes (2005, Theorem 1.8), which we used extensively to derive the results in section 2.2.

**Theorem 28.** *(Hayes, 2005, Theorem 1.8) Let $\mathbf{X} = (X_j)_{j=1}^n$ be a very-weak martingale taking values in $\mathbf{E}$ such that $X_0 = 0\mathbf{1_E}$, and, for every $i \geq 1$, $\|X_j - X_{j-1}\|_2 \leq 1$. Then, for every $a > 0$,*

$$P\left(\|X_j\|_2 \geq a\right) < 2\exp\left(1 - \frac{(a-1)^2}{2j}\right).$$

We then introduce the concepts of filtration and natural filtration. They are crucial for section 3.

**Definition 29.** *Let $(\Omega, \mathcal{F}, P)$ be a probability space and let $I$ be an index set with a total order $\leq$. For every $i \in I$, let $\mathcal{F}_j$ be a sub-$\sigma$-algebra of $\mathcal{F}$. Then, $\mathbf{F} = (\mathcal{F}_j)_{j \in I}$ is a filtration if, for all $k \leq \ell$, $\mathcal{F}_k \subset \mathcal{F}_\ell$.*

**Definition 30.** *Let $(\Omega, \mathcal{F}, P)$ be a probability space and let $I$ be an index set with a total order $\leq$. Let $(S, \Sigma)$ be a measurable space and let $\mathbf{X} : I \times \Omega \to S$ be a stochastic process. Then the natural filtration of $\mathcal{F}$ with respect to $\mathbf{X}$ is defined to be the filtration $\mathbf{F^X} = (\mathcal{F}_j^{\mathbf{X}})_{j \in I}$ given by*

$$\mathcal{F}_j^{\mathbf{X}} := \sigma\left(\left\{X_j^{-1}(A) : j \in I, j \leq i, A \in \Sigma\right\}\right).$$

*That is, $\mathcal{F}_j^{\mathbf{X}}$ is the smallest $\sigma$-algebra on $\Omega$ that contains all pre-images of $\Sigma$-measurable subsets of $S$ for "times" $j$ up to $i$.*

## C Tropical algebra

A detailed introduction to tropical algebra and tropical geometry is provided in Itenberg et al. (2009); Maclagan & Sturmfels (2015). The fundamental element of tropical algebra, the tropical semiring, is given by

$$\mathbb{T} = (\mathbb{R} \cup \{-\infty\}, \oplus, \odot).$$

The two operations $\oplus$ and $\odot$ are called *tropical addition* and *tropical multiplication*, respectively, and are such that $a \oplus b := \max\{a, b\}$ and $a \odot b := a + b$, for all $a, b \in \mathbb{R} \cup \{-\infty\}$. The distributive law holds for tropical addition and multiplication, the identity element of tropical addition is $-\infty$, and the identity element of tropical multiplication is $0$. The tropical semiring is idempotent in the sense that $a \oplus a \oplus \cdots \oplus a = a$, for all $a \in \mathbb{R} \cup \{-\infty\}$. Because of this, there is no tropical subtraction, but tropical division is well defined

$$a \oslash b := a - b,$$

for all $a, b \in \mathbb{R} \cup \{-\infty\}$. Tropical exponentiation is well defined as well, for all $a \in \mathbb{R} \cup \{-\infty\}$, we have that

$$a^{\odot b} := \begin{cases} a \cdot b & \text{if } b \in \mathbb{Z}_+ \\ (-a) \cdot (-b) & \text{if } b \in \mathbb{Z}_- \end{cases}.$$

As we can see, tropical exponentiation is well defined only for integer exponents. We also have that

$$-\infty^{\odot a} := \begin{cases} -\infty & \text{if } a > 0 \\ 0 & \text{if } a = 0 \\ \text{undefined} & \text{if } a < 0 \end{cases}.$$

Notice that the tropical semiring is a (non-Diophantine) abstract prearithmetic, where the partial order is defined on the extended reals (Burgin & Czachor, 2020; Caprio et al., 2021). We can now define tropical polynomials and tropical rational functions.

A *tropical monomial* in $d$ variables $x_1, \ldots, x_d$ is an expression of the form

$$c \odot x_1^{\odot a_1} \odot x_2^{\odot a_2} \odot \cdots \odot x_d^{\odot a_d},$$

where $c \in \mathbb{R} \cup \{-\infty\}$ and $a_1, \ldots, a_d \in \mathbb{N}$. As a notational shorthand, we can write it in multi-index notation as $cx^\alpha$, where $\alpha = (a_1, \ldots, a_d)^\top \in \mathbb{N}^d$ and $x = (x_1, \ldots, x_d)^\top$.

A *tropical polynomial* $f(x) = f(x_1, \ldots, x_d)$ is a finite tropical sum of tropical monomials,

$$f(x) = c_1 x^{\alpha_1} \oplus \cdots \oplus c_r x^{\alpha_r},$$

where $\alpha_i = (a_{i1}, \ldots, a_{id})^\top \in \mathbb{N}^d$ and $c_i \in \mathbb{R} \cup \{-\infty\}$, $i \in \{1, \ldots, r\}$. We assume that $\alpha_i \neq \alpha_j$, for all $i \neq j$.

A *tropical rational function* is a standard difference, that is, a tropical quotient of two tropical polynomials $f(x)$ and $g(x)$,

$$f(x) - g(x) = f(x) \oslash g(x).$$

We denote a tropical rational function by $f \oslash g$, where $f$ and $g$ are tropical polynomial functions. A tropical polynomial $f$ can be seen as a tropical rational function, indeed $f = f \oslash 0$. Hence, any result holding for tropical rational functions hold also for tropical polynomials.

A $d$-variate tropical polynomial $f(x)$ defines a function $f : \mathbb{R}^d \to \mathbb{R}$ that is a convex function, in that taking max and sum of convex functions preserves convexity (Boyd & Vandenberghe, 2004). So, a tropical rational function $f \oslash g : \mathbb{R}^d \to \mathbb{R}$ is a difference of convex function (An & Tao, 2005; Hartman, 1959).

A function $F : \mathbb{R}^d \to \mathbb{R}^p$, $x = (x_1, \ldots, x_d)^\top \mapsto (f_1(x), \ldots, f_p(x))^\top$, is called a *tropical polynomial map* if each $f_i : \mathbb{R}^d \to \mathbb{R}$ is a tropical polynomial, for all $i \in \{1, \ldots, p\}$, and a *tropical rational map* if $f_1(x), \ldots, f_p(x)$ are tropical rational functions.

Tropical polynomials and tropical rational functions are piecewise linear functions. Hence, a tropical rational map is a piecewise linear map and the notion of a linear region applies. A *linear region* of a tropical rational map $F$ is a maximal connected subset of the domain on which $F$ is linear. The number of linear regions of $F$ is denoted by $\mathcal{N}(F)$.

# D  Proofs

*Proof of Theorem 4.* Immediate from Definition 23 and Proposition 24. □

*Proof of Corollary 5.* Immediate from Theorem 4 and equation equation 2. □

*Proof of Corollary 7.* Immediate from Theorem 6. □

*Proof of Proposition 12.* Immediate from Propositions 21 and 22. □

*Proof of Proposition 13.* Let us first assume $\mathbb{E}_{(X,Y)\sim\mathscr{D}}[s(\nu(X))] < c$. By Proposition 12, we have that, for all $t > 0$,

$$P_{(X,Y)\sim\mathscr{D}}\left(s(\nu(X)) \geq \mathbb{E}_{(X,Y)\sim\mathscr{D}}[s(\nu(X))] + t\right) \leq \exp\left(-\frac{2t^2}{(b-a)^2}\right). \tag{24}$$

Then, since $\mathbb{E}_{(X,Y)\sim\mathscr{D}}[s(\nu(X))] < c$, there exists $t_2 > 0$ such that $\mathbb{E}_{(X,Y)\sim\mathscr{D}}[s(\nu(X))] + t_2 = c$. The bound in equation 24 implies that

$$P_{(X,Y)\sim\mathscr{D}}\left(s(\nu(X)) \geq \mathbb{E}_{(X,Y)\sim\mathscr{D}}[s(\nu(X))] + t_2\right) = P_{(X,Y)\sim\mathscr{D}}\left(s(\nu(X)) \geq c\right) \leq \exp\left(-\frac{2t_2{}^2}{(b-a)^2}\right).$$

Assume now that $\mathbb{E}_{(X,Y)\sim\mathscr{D}}[s(\nu(X))] > c$. By Proposition 12, we have that, for all $t > 0$,

$$P_{(X,Y)\sim\mathscr{D}}\left(s(\nu(X)) \leq \mathbb{E}_{(X,Y)\sim\mathscr{D}}[s(\nu(X))] - t\right) \leq \exp\left(-\frac{2t^2}{(b-a)^2}\right). \tag{25}$$

Then, since $\mathbb{E}_{(X,Y)\sim\mathscr{D}}[s(\nu(X))] > c$, there exists $t_1 > 0$ such that $\mathbb{E}_{(X,Y)\sim\mathscr{D}}[s(\nu(X))] - t_1 = c$. The bound in equation 25 implies that

$$P_{(X,Y)\sim\mathscr{D}}\left(s(\nu(X)) \leq \mathbb{E}_{(X,Y)\sim\mathscr{D}}[s(\nu(X))] - t_1\right) = P_{(X,Y)\sim\mathscr{D}}\left(s(\nu(X)) \leq c\right) \leq \exp\left(-\frac{2t_1{}^2}{(b-a)^2}\right).$$

□

*Proof of Proposition 16.* If $\mathscr{S}_j$ and $\mathscr{T}_j$ are bounded for all $j \in \{1, \ldots, n_0\}$, then $F^{(0)}(X)$ and $G^{(0)}(X)$ are bounded. In addition, if $\mathscr{P}$ and $\mathscr{Q}$ are bounded, then $A^{(l)}$ and $b^{(l)}$ are bounded, for all $l \in \{1, \ldots, L\}$. In turn, this entails that $\nu^{(l)}(X)$ is bounded, that is, there exists $\xi_{(l)} \in \mathbb{R}$ such that $\|\nu^{(l)}(X)\|_2 \leq \xi_{(l)}$, for all $l \in \{1, \ldots, L\}$. Then, the fact that equation 3 holds comes immediately from Corollary 5. □

*Proof of Proposition 19.* We prove the claim only for $\mathcal{N}(f)$, as the proof for $\mathcal{N}(g)$ is analogous. Notice that $\mathcal{N}(f) \in \mathbb{N}$, so given our assumption that for some $b_1 > 1$, $\mathcal{N}(f) \leq b_1$ with probability 1, we can write that $1 \leq \mathcal{N}(f) \leq b_1$ with probability 1. Then, by Proposition 22, we have that $\mathcal{N}(f)$ is $SG(\frac{b_1-1}{2})$. So, by Proposition 21, for all $t > 0$,

$$P_{(X,Y)\sim\mathscr{D}}\left(\left|\mathcal{N}(f) - \mathbb{E}_{(X,Y)\sim\mathscr{D}}[\mathcal{N}(f)]\right| \geq t\right) \leq 2\exp\left(-\frac{t^2}{2\frac{(b_1-1)^2}{4}}\right)$$

$$= 2\exp\left(-\frac{2t^2}{(b_1-1)^2}\right),$$

concluding the proof. □

*Proof of Proposition 21.* Let $X$ be $SG(\sigma)$. We have that, for all $s > 0$,

$$P(X - \mathbb{E}(X) \geq t) \leq P\left(e^{s(X-\mathbb{E}(X))} > e^{st}\right) \leq \frac{\mathbb{E}\left(e^{s(X-\mathbb{E}(X))}\right)}{e^{st}},$$

where the first inequality comes from using Markov's inequality, and the second one from using Chernoff's bound. Then, since $X$ is $SG(\sigma)$, we have that

$$P(X - \mathbb{E}(X) \geq t) \leq \exp\left(\frac{s^2\sigma^2}{2} - st\right).$$

The above inequality holds for any $s > 0$ so to make it the tightest possible, we minimize with respect to $s > 0$. In particular, we solve $\phi'(s) = 0$, where $\phi(s) = \frac{s^2\sigma^2}{2} - st$, and we find $\inf_{s>0} \phi(s) = -\frac{t^2}{2\sigma^2}$. We complete the proof by repeating this process for $P(X - \mathbb{E}(X) \leq -t)$. $\qquad\square$

*Proof of Proposition 22.* Without loss of generality, let $\mathbb{E}(X) = 0$. Then, let $P$ denote the probability distribution of $X$. Pick any $\theta \in \mathbb{R}$ and define $\varphi(\theta) := \log \mathbb{E}_P(e^{\theta X})$. Let then $Q_\theta$ be the distribution of $X$ defined by

$$\mathrm{d}Q_\theta(x) := \frac{e^{\theta x}}{\mathbb{E}_P(e^{\theta X})}\mathrm{d}P(x).$$

We have that

$$\varphi'(\theta) = \frac{\mathbb{E}_P(Xe^{\theta X})}{\mathbb{E}_P(e^{\theta X})} = \int x \frac{e^{\theta x}}{\mathbb{E}_P(e^{\theta X})}\mathrm{d}P(x) = \mathbb{E}_{Q_\theta}(X)$$

and that

$$\varphi''(\theta) = \frac{\mathbb{E}_P(X^2 e^{\theta X})}{\mathbb{E}_P(e^{\theta X})} - \frac{\mathbb{E}_P(Xe^{\theta X})^2}{\mathbb{E}_P(e^{\theta X})^2}$$
$$= E_{Q_\theta}(X^2) - E_{Q_\theta}(X)^2 = \mathbb{V}_{Q_\theta}(X),$$

where $\mathbb{V}$ denotes the variance operator. Now, by Popoviciu's inequality we have that $\mathbb{V}_{Q_\theta}(X) \leq \frac{(b-a)^2}{4}$. Then, by the fundamental theorem of calculus, we have that

$$\varphi(\theta) = \int_0^\theta \int_0^\mu \varphi''(\rho) \, \mathrm{d}\rho \, \mathrm{d}\mu \leq \frac{(b-a)^2}{4}\frac{\theta^2}{2}, \tag{26}$$

using $\varphi(0) = \log 1 = 0$ and $\varphi'(0) = \mathbb{E}_{Q_0}(X) = E_P(X) = 0$. The proof is concluded by exponentiating both sides of the inequality in equation 26. $\qquad\square$

