# OpenReview forum: "Concentration inequalities and optimal number of layers for stochastic deep neural networks"
_TMLR — Rejected by TMLR_

### Review · Reviewer_GwBG · 2022-07-27

**Summary Of Contributions:**

In this work, the authors derive sub-Gaussian concentration inequalities for the outputs of stochastic deep neural networks (SDNNs). They show that these concentration inequalities imply concentration in the score/loss function as well. The analysis uses properties of martingales in order to build up the concentration results layer-by-layer.

The authors also provide an analysis of the optimal depth of a SDNN, subject to a specific regularization which penalizes large depth networks. Finally, they apply their concentration theorems to a basic ReLu based SDNN, and analyze the number of connected negative regions.

**Requested Changes:**

Critical changes:

For a more general audience, it would be helpful to describe stochastic NN in the introduction - perhaps with a simple, explicit example. This makes it easier to understand the setting, and will make the paper more self-contained (which would overall strengthen the submission).

As a stylistic note, sub-Gaussian looks better than subGaussian. My secondary choice would be subgaussian.

The labelling of the first two theorems is confusing to me. For one, they are somewhat informal/imprecise; this is not an issue per-se, but they should be marked as such. Secondly, while reading the text, I expected them to recur in their more formal versions. However, it seems that the results alluded to are split up over the remaining theorems and propositions - e.g. Corollary 4 seems to have much of the content of Theorem 1, and Propositions 11 and 12 seem to correspond to Theorem 2. The paper would be made much clearer if:
* The theorems were explicitly noted to be informal.
* The formal versions of the theorem were explicitly present and referenced with the same theorem numbers.
Otherwise, it looks as if those theorems have not been proven!

There are two uses of Y in corollary 6; one is the responses in the data distribution, and the other is the sequence of mean-centered outputs. One of these should be changed to avoid confusion/overload of terminology.

Remark 10 seems false to me; as the number of layers increases, so does the bound (as the negative numerator of the exponential in the bound decreases with l, the number of layers).

In Section 3, more discussion of the chosen loss function is warranted. In particular, it would be helpful to explain the choice of powers for the loss term and the l term. In addition, it is not clear to me that the stated results are helpful for optimization in practice. In particular, it seems as if the loss+layer penalty could be non-monotonic - meaning that all possible values of the number of layers would have to be explored in order to find the optimum. A specific example would convince the reader (and myself) of the usefulness of the results.

The proof of Proposition 18 was not clear to me; it would be helpful to more explicitly spell out which of the previous results is used to derive it.

Repeated word in the line "Equation equation 7 tells us".

Optional changes:

Since Hayes 2005 theorem 1.8 seems to be important for the proofs of Section 2, summarizing it in either the main text or one of the appendices will improve the self-contained nature of the paper.

Condense section 4 and move most of the details to the appendix. Give more explicit explanation for how the previous theorems lead into propositions 17 and 18.

**Strengths And Weaknesses:**

The concentration inequalities in Section 2.1 and 2.2 are simple, elegant, and seem correct. In addition, the instinct of the authors to make the paper as self-contained as possible is a big strength - it allows for a wider audience to absorb the results, and consult the related literature as needed.

Section 2.3 is independent of the previous two sections, and seems to come from trivial applications of the definitions of sub-Gaussianity. I don't believe it adds much value to the work, or is of interest to the leadership of TMLR.

Section 3 provides, in principle, a method for computing the optimal depth of an SDNN given a specific cost function which takes into account a combination of the loss and the number of layers. However, there is no motivation or explanation for this specific loss function. In addition, it seems that the analysis does not deal with the case where the loss function is non-monotonic in the number of layers - the backward induction will stop at any local maximum of the score function. In general, the potential non-monotonicity suggests that there is no general efficient method to find the optimum number of layers.

Overall, Section 4 felt long/distracting from the main content of the paper. The development of the tropical algebra is carried out in Zheng 2018 (as the authors themselves identify). The original results are Propositions 17 and 18, which give concentration inequalities. It is not clear to me if this section overall is of interest to the readership of TMLR. However, this is also a research area I am less familiar with, and if other reviewers with more experience in the area think the section is important, I am happy to defer to their expertise.

Regarding the results of section 4: it is stated that the assumption "the weight matrices A(1), . . . , A(L) are integer-valued" is not an issue due to the possibility of rational approximation. However in many practical settings, there can be issues with the limitations of machine precision. It is not clear how this affects the utility of the proof.

Overall I am very confident in my assessment of the correctness of the results, and somewhat confident in my ability to judge their interest to the readership of TMLR given the other related work.

---

> ### Author Response · Authors · 2022-08-24
> **Response to reviewer GwBG**
>
> We appreciated the constructive comments made to our work. To answer them,
> •	we described stochastic NN in the introduction – also giving the example of Bayesian Neural Networks
> •	we now write sub-Gaussian in place of subGaussian
> •	we explicitly stated that propositions 1 and 2 are informal, and where the formal version is. In the main portion of the paper, then, we referenced propositions 1 and 2 when the formal version is introduced
> •	we substituted Y (used for the sequence of mean-centered outputs) in corollary 6 with Z
> •	we especially appreciated the concern about remark 10. It turns out that in the whole section 2.2 the bounds increase with the number of layers. This means that the results we give in that section better suit shallow networks, as the inequalities become looser as the number of layers increases
> •	we especially appreciated the concern about the loss function in section 3. We completely rewrote the initial part of section 3, which now addresses all the raised concerns. As a result, the section is greatly improved
> •	we wrote the proof to proposition 18 explicitly
> •	we corrected the noted typo
> •	we summarized Hayes 2005, Theorem 1.8 in the appendix
> •	we condensed section 4 and moved most of the details to the appendix. We also gave more explicit explanation for how the previous theorems and the results in Zhang et al. lead into propositions 17 and 18 in section 4.3

---

### Review · Reviewer_6RJf · 2022-07-28

**Summary Of Contributions:**

This work deals with generic stochastic deep neural networks (SDNN). The work first gives the concentration inequalities of the hidden states. Section 1.2 summarizes the contribution well. In particular, i’d like to highlight the following:
- Theorem 8 provides a sufficient condition for a network to be a weak martingale.
- Corollary 6 shows that if a NN is a weak martingale, then under a weak assumption, l-th layer hidden states concentrate around the expected value.
- Propositions 11 and 12 analyze the expected decision boundary and the expected classifier.
- Theorem 14: Assuming the network has a constant width p, to deal with the accuracy (for the classifier) vs. cost (computational cost), propose the number of layers that’s optimal for the trade-off between accuracy and cost.
- The analysis is then applied onto an SDNN with ReLU activation based on Zhang et al. (2018) – the set of NNs with ReLU is equivalent to a set of tropical rational maps; also see Figure 1. In particular, the SDNN example has stochasticity in the initialization of the network, and the parameters are aleatory.  The authors check that the l-th layer hidden states are bounded, and the norm is bounded. Given that the condition in Corollary 4 is satisfied, the authors go on deriving the bounds for the number of positive/negative regions of the network.


**Requested Changes:**

- Elaborate the comparison to previous work, as discussed above.
- Elaborate the significance of using the probabilistic framework for Zhang et al.

**Strengths And Weaknesses:**

Writing is quite clear (except for a few typos here and there: the third to the last line on page 2, EBD should be EDB in the first paragraph of Section 1.2, etc.).

The work proposes the optimal number of layers that strike a good balance between accuracy and cost (see Theorem 14) for a generic stochastic deep neural network, by an optimal stopping procedure.

I checked the proofs all the way till Theorem 14. I skimmed through the rest of the proofs for the real examples. The logic is clear. I have no complaints so far.


---


In general, it would be helpful to clarify the differences among your framework, Ost and Reynaud-Bouret (2020; https://projecteuclid.org/journals/annales-de-linstitut-henri-poincare-probabilites-et-statistiques/volume-56/issue-4/Sparse-spacetime-models-Concentration-inequalities-and-Lasso/10.1214/19-AIHP1042.short), and Garnier and Langhendries (2022; https://projecteuclid.org/journals/electronic-journal-of-statistics/volume-16/issue-1/Concentration-inequalities-for-non-causal-random-fields/10.1214/22-EJS1992.full). To be fair, the authors already cited the work. I agree that their work deals with specific types of networks, but it will be great if the authors can explain in more detail why the authors’ framework requires less mathematical structure, whether the authors’ framework is inspired by the previous frameworks in any way, and why readers / some of the TMLR audience members would be interested in the framework.

It’s not too clear to me why using the probabilistic framework to study Zhang et al.’s (https://arxiv.org/abs/1805.07091) framework (of connecting NNs with ReLUs and tropical rational maps) is significant – what future directions would the probabilistic framework enable?

---

> ### Author Response · Authors · 2022-08-24
> **Response to reviewer 6RJf**
>
> We appreciated the constructive comments made to our work. To answer them,
> •	we elaborated on the difference between our work, Ost and Reynaud-Bouret (2020), and Garnier and Langhendries (2022). We explained why our framework requires less mathematical structure, which papers inspired some of the techniques we used for our results, and why some TMLR audience should be interested in our work in section 1.2
> •	we elaborated on the significance of finding probabilistic results in the framework of Zhang et al., and what future directions do these results enable in section 1.2

---

### Review · Reviewer_6t6b · 2022-08-11

**Summary Of Contributions:**

The paper makes boundedness and martingale assumptions on the outputs of the layers. Several concentration results follow immediately given these assumptions from known results in probability. These results are stated as theorems, and form the basis of the contributions by the authors.


**Broader Impact Concerns:**

None.

**Requested Changes:**

I see no nontrivial changes that are of any utility.

**Strengths And Weaknesses:**

To summarize, nearly all of the theorems are trivial (either known results, like proposition 11, follow immediately from the definitions, like theorem 3, or are straightforward consequences). The paper reads as an exercise in probability, naming the variables using machine learning terminology, and restating the definitions and known results. I do not see how this paper advances our knowledge of machine learning at all.

The term concentration is used throughout, but this term is used in a strange way in my opinion. In reality, the paper gives a number of tail bounds. One usually reserves the term concentration for when the tail probability vanishes over some natural sequences (such as increasing data, depth, etc.). There is no concentration going on in this latter (more useful) sense.

The applications seem to be manufactured and are of no interest to machine learning in their present form.

---

> ### Author Response · Authors · 2022-08-24
> **Response to reviewer 6t6b**
>
> We appreciated the honest comments made to our work. In light of them, we added a line at the beginning of section 2 where we point out how the results in that section are direct applications of standard tools in high-dimensional probability.
> We disagree that this paper does not advance the knowledge of machine learning at all, for two main reasons.
> •	The first one is that section 3 we portrayed a generic SDNN as a stochastic process. This allowed us to use a backward induction approach to an optimal stopping problem to find its optimal number of layers. We find this take on SDNNs to be new and important for the field, as more properties can be found in the future by viewing SDNNs as stochastic processes.
> •	The second one is that there is a non-negligible portion of the machine learning community that might be not aware of the results in section 2. In addition, the fact that we do not require any particular architecture or structure for the neural network makes these results appealing, as they hold for all possible SDNNs.
> We agree that the results in section 2.2 are actually tail bounds, and we added a footnote to highlight this fact. This means that the results we give in that section better suit shallow networks, as the inequalities become looser as the number of layers increases.
> Finally, we disagree on the fact that the applications appear manufactured. We consider the stochastic version of a widely known and used deep neural network, namely the feedforward NN with ReLU activation (FNNRA) and show that the results that we find in general (section 2.1) are easily applicable to a stochastic FNNRA.
> We hope that the new version of the paper will be met more favorably by the reviewer.

---

### Decision · Action_Editors · 2022-09-20

**Recommendation:** Reject

**Comment:**

This paper investigates stochastic deep neural networks and concentration and martingale inequalities for the output of the hidden layers. The paper was improved in other regards during rebuttal period such as comparison to Ost and Reynaud-Bouret (2020), and Garnier and Langhendries (2022). Although the results are correct and cleanly demonstrated and do not depend on the architecture type, the authors could not convince the reviewers that the results are of interest to TMLR community mainly because the reviewers believe the results are straightforward and trivial(by reusing the ML notations in already existing frameworks) and provide no application in real scenarios as well as only being about shallow networks. The vote between reviewers is unfortunately unanimous and the authors need to resolve this significant issue before the paper can be ready to acceptance to TMLR.